# Cancer-Associated-Fibroblast-Mediated Paracrine and Autocrine SDF-1/CXCR4 Signaling Promotes Stemness and Aggressiveness of Colorectal Cancers

**DOI:** 10.3390/cells13161334

**Published:** 2024-08-12

**Authors:** Chao-Yang Chen, Shih-Hsien Yang, Ping-Ying Chang, Su-Feng Chen, Shin Nieh, Wen-Yen Huang, Yu-Chun Lin, Oscar Kuang-Sheng Lee

**Affiliations:** 1Division of Colon and Rectal Surgery, Tri-Service General Hospital, National Defense Medical Center, Taipei 11490, Taiwan; cartilage88@gmail.com; 2Institute of Clinical Medicine, National Yang Ming Chiao Tung University, Taipei 11221, Taiwan; 3Graduate Institute of Medical Sciences, National Defense Medical Center, Taipei 11490, Taiwan; thero0823@gmail.com; 4Office of General Affairs and Occupational Safety, National Defense Medical Center, Taipei 11490, Taiwan; 5Division of Hematology-Oncology, Department of Internal Medicine, Tri-Service General Hospital, National Defense Medical Center, Taipei 11490, Taiwan; max-chang@ndmctsgh.edu.tw; 6Department of Dentistry, School of Dentistry, China Medical University, Taichung 40433, Taiwan; csf@mail.cmu.edu.tw; 7Department of Pathology, Tri-Service General Hospital, National Defense Medical Center, Taipei 11490, Taiwan; ns1014@mail.ndmctsgh.edu.tw; 8Department of Radiation Oncology, Tri-Service General Hospital, National Defense Medical Center, Taipei 11490, Taiwan; hwyyi@mail.ndmctsgh.edu.tw; 9Stem Cell Research Center, National Yang Ming Chiao Tung University, Taipei 11211, Taiwan; 10Department of Orthopedics, China Medical University Hospital, Taichung 40402, Taiwan; 11Center for Translational Genomics & Regenerative Medicine Research, China Medical University Hospital, Taichung 40402, Taiwan

**Keywords:** cancer-associated fibroblasts (CAFs), stromal-derived factor-1 (SDF-1), C-X-C motif chemokine receptor 4 (CXCR4), tumor microenvironment (TME), cancer stem cells (CSCs)

## Abstract

Colorectal cancer (CRC) is a leading cause of cancer mortality worldwide, and cancer-associated fibroblasts (CAFs) play a major role in the tumor microenvironment (TME), which facilitates the progression of CRC. It is critical to understand how CAFs promote the progression of CRC for the development of novel therapeutic approaches. The purpose of this study was to understand how CAF-derived stromal-derived factor-1 (SDF-1) and its interactions with the corresponding C-X-C motif chemokine receptor 4 (CXCR4) promote CRC progression. Our study focused on their roles in promoting tumor cell migration and invasion and their effects on the characteristics of cancer stem cells (CSCs), which ultimately impact patient outcomes. Here, using in vivo approaches and clinical histological samples, we analyzed the influence of secreted SDF-1 on CRC progression, especially in terms of tumor cell behavior and stemness. We demonstrated that CAF-secreted SDF-1 significantly enhanced CRC cell migration and invasion through paracrine signaling. In addition, the overexpression of SDF-1 in CRC cell lines HT29 and HCT-116 triggered these cells to generate autocrine SDF-1 signaling, which further enhanced their CSC characteristics, including those of migration, invasion, and spheroid formation. An immunohistochemical study showed a close relationship between SDF-1 and CXCR4 expression in CRC tissue, and this significantly affected patient outcomes. The administration of AMD3100, an inhibitor of CXCR4, reversed the entire phenomenon. Our results strongly suggest that targeting this signaling axis in CRC is a feasible approach to attenuating tumor progression, and it may, therefore, serve as an alternative treatment method to improve the prognosis of patients with CRC, especially those with advanced, recurrent, or metastatic CRC following standard therapy.

## 1. Introduction

Despite drastic improvements in the treatment of colorectal cancer (CRC) through surgical advances and innovations in adjuvant therapy, including chemo- and radiotherapy, about one-third of patients who are surgically treated for CRC will develop recurrence and/or distant metastases—most commonly in the liver and lungs, two major organs that contribute to CRC-related mortality [1,2,3]. In addition, the so-called tumor microenvironment (TME), the ecosystem surrounding tumor cells, which also contains immune cells, fibroblasts, endothelial cells, and stromal proteins, significantly influences the CRC cell behavior at the primary tumor site [4,5]. During tumor progression, there is a coevolution of the “seed”, i.e., the cancer cell, with the surrounding tumor microenvironment—the “soil”. There is extensive crosstalk among these different cell types—together, they help promote the processes of tumor growth and development [6,7]. It has been suggested that targeting stromal events could enhance the efficacies of existing therapeutics and diminish metastatic spread [8,9,10].

Fibroblasts are spindle-shaped cells that secrete collagen and have a cytoplasm that is unusually rich in rough endoplasmic reticula. They synthesize the extracellular matrix of connective tissues and play an active and crucial role in wound healing and cancer [11,12,13]. Cancer-associated fibroblasts (CAFs) are uniquely versatile, functioning as paracrine factories or autocrine mediators. They control the trophic influence on cancer cells through stimulation [14]. Fibroblasts are termed cancer-linked fibroblasts when they are associated with cancer stem cells (CSCs) and facilitate tumor development. They promote tumorigenesis by adjusting the microenvironment, physically joining cancer cells, depositing extracellular matrix components, participating in angiogenesis, and providing an outward avenue for metastasis [15,16,17]. CAFs stimulate metastasis through extracellular matrix reconstruction, cytokine discharge, and participating in the epithelial–mesenchymal shift in multiple forms of cancer, including CRC [18,19,20,21,22,23,24].

Stromal-derived factor-1 (SDF-1) is known to function as a chemotactic factor for lymphocytes, dendritic cells, and monocytes. Its corresponding receptor, C-X-C motif chemokine receptor 4 (CXCR4), is a seven-span trans-membrane G-protein-coupled receptor with 352 amino acids that selectively binds the ligand SDF-1 [25]. The recent literature reported an increased level of the chemokine SDF-1 and its receptor in patients with colon cancers, and the SDF-1/CXCR4 axis is considered a valuable marker of cancer metastasis [26,27,28,29,30,31]. High levels of SDF-1 were found to be mediated by CAFs in 36.8% of patients with CRC [32,33,34].

The purpose of this study was to elucidate the role of CAFs in promoting the aggressiveness of CRC through the modulation of the SDF-1/CXCR4 signaling pathway via paracrine and autocrine effects. Understanding the driving force of tumor progression and the relationship between cancer cells and the TME can be fundamental in developing innovative therapeutic strategies for a better and definitive response in patients with CRC.

## 2. Materials and Methods

### 2.1. Isolation, Extraction, and Identification of CAFs and NFs

Fresh colon cancer samples were collected from patients who were pathologically diagnosed with CRC at the Tri-Service General Hospital. Informed consent was obtained from all participants, and approval was obtained from the Human Research Ethics Committee. The samples were sectioned into blocks that were approximately 1 to 2 mm in diameter and then digested using trypsin and 0.5% collagenase, followed by filtration through a strainer to isolate fibroblasts (NFs and CAFs) from the cell suspension. The isolated cells were cultured in Dulbecco Modified Eagle Medium (DMEM) and Ham’s Nutrient Mixture F12 supplemented with 10% fetal bovine serum (FBS) and 1% penicillin–streptomycin (Gibco, Life Technologies, Carlsbad, CA, USA), and they were maintained at 37 °C in a 5% CO_2_ atmosphere; the medium was replaced after 24 h. CAFs were identified based on their morphological characteristics and immunohistochemical staining for α-smooth muscle actin (α-SMA) and vimentin.

### 2.2. RNA Extraction, Quality Assessment, and Microarray Analysis

Total RNA was extracted from both the treated and control cells using the Bio-Ray (Pingtung, Taiwan) Aurum^TM^ Total RNA Mini Kit while adhering strictly to the manufacturer’s instructions. The integrity and concentration of the extracted RNA were assessed using a Bioanalyzer system, with only samples achieving an RNA integrity exceeding 8.0 being processed for further analysis. The microarray experiments were conducted using an Affymatrix chip that was specifically designed for human gene expression analysis. Procedures for sample labeling, microarray hybridization, and washing were performed according to the manufacturer’s protocols, and each condition was analyzed in triplicate to ensure reproducibility. For data normalization and analysis, raw data were extracted and normalized to correct any systemic variations. Genes were considered significantly differentially expressed based on a fold-change threshold of log2 (fold change) > 1 and a *p*-value of <0.05, ensuring robustness in the findings.

### 2.3. Cell Cultures and 3D Organotypic Culture System

HT-29 and HCT-116 cell lines originating from human colorectal adenocarcinomas were obtained from the American Type Culture Collection (ATCC) and cultured according to standard protocols. An organotypic culture system was used, as described in our previous study [35]. To grow organotypic 3D cultures, a mixture was prepared by combining eight volumes of collagen I/Matrigel (in a ratio of 1:1) with one volume of 10× DMEM and one volume of FBS containing fibroblasts (at a concentration of 5 × 10^6^ cells/per well). This gel mixture was then dispensed into a 12 mm Millicell insert (Millipore, Bedford, MA, USA), which was inserted into a six-well culture plate. The gel was allowed to solidify at a temperature of 37 °C for a duration of 24 h. Following this, HT-29 and HCT-116 cells (at a density of 2 × 10^5^ cells/per well) were placed on top of the gel mixture. After an incubation period of 24 h, the medium on the surface was removed to expose the cancer cells to air. The gel was then nourished from below with complete medium, which was replaced daily. After a period of 14 days, the cultivated tissue was fixed and embedded in paraffin for histological examination.

### 2.4. Viral Production and Infection of Target Cells

The SDF-1 coding sequence in its entirety was subsequently excised from the pMSCV vector and subcloned into the mammalian expression vector pMSCV-SDF-1. Validation of the ultimate construct was carried out through sequencing. Ampicillin was utilized for the selection of positive bacterial clones carrying the plasmid, which were then expanded in bacterial culture. The purification of all plasmids was conducted by employing the Endo-free Plasmid Mini Kit II (OMEGA). Prior to transfection, the HT-29 and HCT-116 cells were seeded in six-well plates (Corning, Lowell, MA, USA), cultured in a complete growth medium until reaching 80% confluency, and subsequently maintained in a medium devoid of FBS for 12–16 h. Transfection was accomplished by employing 1 μg (per dish) of either p pMSCV or p pMSCV–SDF-1 and Turbofect (Thermo Fisher Scientific Inc., Branchburg, NJ, USA) reagent according to the manufacturer’s guidelines. At 12 h post-transfection, the media were supplemented with 2 μg/mL puromycin (Invitrogen, Waltham, MA, USA). Following a two-week incubation period, clones were selected and cultured in six-well plates until reaching confluence. Stable HT-29-pMSCV, HCT-116-pMSCV (HT-29 and HCT-116 cells transfected with an empty vector (HT29-Ctrl and HCT-116-Ctrl)), HT-29-pMSCV-SDF-1, and NPC204-pMSCV-SDF-1 (HT-29 and HCT-116 cells transfected with a vector encoding human SDF-1 (HT-29-SDF-1 and HCT-116-SDF-1)) cell lines were obtained and subcultured. The plasmid and SDF-1 sequence are documented in Appendix A.

### 2.5. In Vitro Migration and Invasion Assay

An in vitro migration/invasion assay was conducted in accordance with a previously documented protocol [35]. Transwell inserts (pore size, 8 µm) and lower wells were coated with 15 µg/mL collagen type I, incubated for 1 h at 37 °C, and blocked overnight with phosphate-buffered saline (PBS) containing 1% bovine serum albumin at 4 °C. Subsequently, the blocking buffer was removed and the lower wells were loaded with 300 µL of 50 ng/mL recombinant SDF-1 (rSDF-1) protein (R&D Systems) in serum-free DMEM or serum-free DMEM only (negative control). pMSCV SDF-1 cells were serum-starved overnight and harvested with enzyme-free cell detaching buffer. The cells were incubated with 25 µg/mL AMD3100 in serum-free DMEM or serum-free DMEM only for 2 h at 37 °C. Inserts were loaded with 12 × 10^4^ cells at 150 µL per condition and were allowed to migrate for 12 h at 37 °C. For the invasion assay, a Matrigel/medium (1:2) mixture was placed on the membrane of the upper chamber before seeding the cancer cells. After 12 h of incubation for the migration assay or 24 h of incubation for the invasion assay, non-migratory/invaded cells were removed with a cotton swab wetted in PBS, and migratory or invaded cells were fixed in 4% formaldehyde and stained with hematoxylin at room temperature. The number of migratory cells was calculated by counting cells from five fields of view per slide with 40× magnification while using a counting grid. The obtained data, which were derived from three independent experiments that were performed in triplicate, are presented as the mean ± SD.

### 2.6. Western Blotting

Western blotting was executed in accordance with a standardized protocol, as previously outlined [36]. Briefly, cells were harvested and lysed with RIPA buffer (50 mM Tris, 150 mM NaCl, 0.1% SDS, 0.5% deoxycholate, and 1% NP-40). Protein extracts were subjected to SDS–PAGE analysis. The proteins were transferred to membranes. The membranes were blocked with 5% nonfat milk followed by antibody hybridization, and the signals were then observed on X-ray film and quantified using Image J for Western blotting analysis. The antibodies and primers utilized are documented in Appendix A.

### 2.7. Quantitative Real-Time PCR

For RNA extraction, total RNA was derived from cultivated cells by employing TRIzol (Invitrogen Life Technologies), and 1 μg of RNA was employed for cDNA synthesis. Quantitative real-time PCR (qRT-PCR) was conducted to assess gene expression using the StepOnePlus real-time PCR system (Applied Biosystems, Waltham, MA, USA). The primers utilized are documented in Appendix A.

### 2.8. Enzyme-Linked Immunosorbent Assay

The media from the NF-CM, CAF-CM, HT-29, and HCT-116 cell lines were sampled at 48 h after plating in 24-well plates and centrifuged to remove cell debris. The SDF-1 and CXCR4 levels in the media were assayed with the Quantikine Human SDF1 and CXCR4 Immunoassay Kit (R&D Systems, Abingdon, UK) according to the manufacturer’s instructions. The measured levels were expressed as picograms of SDF-1 and CXCR4 per 1 mg of protein in the cell lysate. ELISA assays were performed in duplicate three separate times, and the data are expressed as the mean ± SD. The antibodies and primers utilized are documented in Appendix A.

### 2.9. Flow Cytometry

A single-cell suspension containing 1 × 10^6^ trypsinized cells and spheres was resuspended in 1 mL of phosphate-buffered saline (PBS) and stained with fluorescent conjugated antibodies against CD133, CD44, CD10, and GPR77 (MAB10254, R&D systems, Minneapolis, MN, USA) for 30 min. After labeling, the cells were washed three times with PBS and subsequently stained with a fluorescein isothiocyanate (FITC)- or PE-labeled secondary antibody for 30 min in the dark. The cells were analyzed using a flow cytometer (FACSCalibur; Epics Elite; Coulter Electronics, Mijdrecht, The Netherlands). Data analysis was performed using Kaluza C analysis software (Beckman Coulter Nederland BV, Woerden, The Netherlands). The antibodies and primers utilized are documented in Appendix A.

### 2.10. Xenograft Model

An in vivo investigation of tumorigenicity was conducted in compliance with the regulations of the local ethics committee, which has received full accreditation from the Association for Assessment and Accreditation of Laboratory Animal Care at the National Defense Medical Center (IACUC-15-036). BALB/c nude mice were housed in conditions of 18–26 °C and 30–70% humidity, and they were kept in cages with an individual air supply while following a 12 h dark/12 h light cycle for a period of 7 days prior to the xenograft injection. The injection of parental HT-29-Ctrl/HT29-SDF-1 or HCT-116-Ctrl/HCT-116-SDF-1 cells was carried out when the mice were 5 weeks of age. A volume of 100 μL of cell suspension was subcutaneously injected into each mouse, and it contained varying cell quantities of 1 × 10^6^, 1 × 10^5^, or 1 × 10^4^. Tumors became apparent 7 days post-injection, with the tumor size being assessed and recorded on a weekly basis using the formula (length × width^2^)/2. At 30 days post-inoculation, either subcutaneously or orthotopically, the mice were humanely euthanized under anesthesia. The tumor volume was assessed, and the tumors were formalin-fixed and embedded in paraffin. Sections were stained immunohistochemically with an antibody and underwent quantitative image analysis.

### 2.11. Sphere Culture

Cells were cultivated in 10 cm plastic dishes with a thin agarose film applied to establish a non-adherent setting. The cells were introduced at a concentration of 5 × 10^4^ viable cells per 10 cm dish, and the culture solution was renewed every other day until the development of spheres, in accordance with prior descriptions [24,36,37].

### 2.12. Clinical Tissue Collection with Immunohistochemical Staining

Archival representative tissue specimens were obtained from the Tri-Service General Hospital in Taiwan between 2013 and 2022. These specimens included primary tumors from both early and advanced CRC patients and from patients diagnosed with terminal CRC with evidence of liver involvement determined via histopathological confirmation. These specimens consisted of four groups and included 10 normal colorectal tissue specimens, 10 cases of early CRC patients, 10 cases of advanced CRC patients, and 10 cases of CRC patients with liver metastasis. Immunohistochemistry was conducted on paraffin-embedded sections of the CRC specimens. The detection of staining was achieved through the use of an Envision detection system (peroxidase/DAB+, rabbit/mouse, Dako Cytomation). Each tissue specimen was given a unique numerical code. Institutional review board approval was not necessary in accordance with Dutch law (TSGHIRB No. 2-103-05-158). For immunohistochemistry, a negative control specimen without a primary antibody was employed, while a human retina with immunoreactivity served as a positive control. The intensities of SDF-1 and CXCR4 immunoreactivity in the tumor cells were categorized into four groups: 0 (no staining), 1 (weak staining), 2 (moderate staining), and 3 (strongest intensity). The staining results were evaluated independently by two pathologists who were unaware of the patients’ clinical information. Any discrepancies between the pathologists were resolved through consensus. To evaluate the diagnostic accuracy of SDF-1 and CXCR4 staining intensity, sensitivity and specificity analyses were performed. 

Construction of Contingency Table: A 2 × 2 contingency table was constructed to compare the staining results with the clinical outcomes:

True Positives (TP): Cases where the staining correctly identified the clinical outcome as positive;

False Positives (FP): Cases where the staining incorrectly identified the clinical outcome as positive;

True Negatives (TN): Cases where the staining correctly identified the clinical outcome as negative;

False Negatives (FN): Cases where the staining incorrectly identified the clinical outcome as negative;

Calculation of Sensitivity and Specificity:

Sensitivity was calculated as the proportion of true positive cases correctly identified by the staining:$$Sensitivity=\frac{TP}{TP+FN}$$

Specificity was calculated as the proportion of true negative cases correctly identified by the staining:$$Specificity=\frac{TN}{TN+FP}$$

This methodology allowed us to assess the potential of SDF-1 and CXCR4 staining intensity as diagnostic markers and to determine their effectiveness in identifying relevant clinical outcomes. The staining results were evaluated independently by two pathologists who were blinded to the patients’ clinical information.

### 2.13. Statistical Analysis

An independent Student’s *t*-test or ANOVA was employed to compare continuous variables between groups, while the χ^2^ test was utilized for dichotomous variables. The level of statistical significance was set to *p* < 0.05. All statistical analyses were performed using SPSS version 20 (SPSS Inc., Chicago, IL, USA).

## 3. Results

### 3.1. SDF-1 Was Markedly Increased in Cancer-Associated Fibroblasts

The heterogeneity within the CAF population includes diverse subtypes, such as myofibroblastic, inflammatory, and immunosuppressive fibroblasts [37]. Initially, we employed flow cytometry to identify fibroblasts derived from clinical samples and specifically targeted the markers CD10 and GPR77, which are prevalent in active CAFs. This method underscored the critical need to understand the complex dynamics of the tumor microenvironment, enriching oncology research by shedding light on the nuanced roles that these cells play in cancer progression (Figure 1A).

Subsequently, we conducted a detailed morphological examination of CAFs, noting their elongated spindle or stellate shapes, thin cytoplasmic extensions, and prominent actin stress fibers. Through immunofluorescent staining, we quantified the expression levels of α-SMA and vimentin in CAFs in comparison with those in NFs, observing a marked elevation in CAFs (Figure 1B). This morphological and protein expression analysis highlighted the transformation of fibroblasts within the tumor microenvironment.

To delve deeper into the molecular mechanisms underlying the facilitation of cancer invasion and metastasis by CAFs, we utilized a gene expression microarray. This analysis targeted a select group of secreted proteins, including chemokines and cytokine receptors, identifying significant alterations (at least a two-fold increase) in their expression in CAFs compared with that in NFs (Figure 1C,D). Additionally, qRT-PCR was employed to measure the expression levels of vimentin, fibroblast activation protein, α-SMA, SDF-1, and CXCR4 in CAFs isolated from the CRC patients’ tissues. Our results indicated a substantial elevation in both the mRNA and protein levels of α-SMA, SDF-1, and CXCR4 in CAFs relative to NFs, substantiating their pivotal role in enhancing tumorigenic capabilities (Figure 1E).

### 3.2. CAFs Induce Aggressive Phenotypes of CRC Cells

Fibroblasts play a pivotal role in investigating stromal–cancer crosstalk at the molecular level, which is crucial for the remodeling of the epithelial–mesenchymal transition (EMT). Therefore, we identified CAFs as a key component of our model, given their role in facilitating the SDF-1-induced migration of cancer cells. Underpinning our hypothesis that CAFs upregulate SDF-1 expression to promote SDF-1-induced migration in colon cancer, we incubated CAFs derived from patients’ colon cancer tissues in an ELISA assay. This assay measured the levels of SDF-1 secreted from CAFs and NFs by collecting conditioned media and assessing the SDF-1 expression (Figure 2A). 

Notably, we confirmed that the NF group exhibited a negligible mRNA expression of SDF-1 and CXCR4, which were at minimal to absent levels, as depicted in Figure 2B. There was a substantially higher mRNA and protein expression in CAFs, emphasizing the differential roles of fibroblast subtypes in the tumor microenvironment. These results indicated that CAFs had significantly upregulated the mRNA and protein levels of SDF-1 compared with the levels in other groups, as illustrated in Figure 2A,B.

Furthermore, with CXCR4 having been identified as a receptor for SDF-1 that drives EMT gene expression in CRC cells, it was imperative to explore its role in modulating EMT gene expression. This investigation necessitates further studies to elucidate the mechanisms by which CXCR4 influences EMT gene expression in CRC. To probe the expression of EMT genes regulated by CXCR4 in CRC cells and to determine the role of the CAF-conditioned medium (CAF-CM) containing secreting factors that influence the TME, we assessed how CAF-CM enhances the motility of CRC cells. 

This enhancement was evidenced by increased levels of SDF-1, CXCR4, and EMT-related proteins, such as Twist-1, Snail, and Vimentin, in the CRC cell lines HT-29 and HCT-116 after exposure to CAF-CM, as shown in Figure 2C,D. Moreover, Transwell migration assays were performed to assess the motility of CRC cells in response to conditioned media from NFs and CAFs. The panels display the stained cells that migrated through the membrane, quantifying the enhanced migratory capacity of CRC cells exposed to CAF-conditioned media. Then, invasion assays were used to demonstrate the invasive potential of CRC cells under the influence of factors secreted by NFs and CAFs. The data showed a significant increase in invasion rates for cells treated with CAF-conditioned media, highlighting the role of CAF-derived factors in promoting CRC cells’ invasiveness. The comparative analysis of NF-CM and CAF-CM, as shown in Figure 2E, highlighted the potential of some factors to induce CRC cells’ mobility.

### 3.3. SDF-1 Expression in CAFs Promotes CRC Cell Migration and Invasion through Autocrine and Paracrine Signaling

It has been reported that the binding of SDF-1 to the chemotaxis receptor CXCR4 promotes homing signal transduction, playing a pivotal role in tumor invasion, recurrence, and distant metastasis [35]. To elucidate the effects of SDF-1-mediated promotion on CRC cells, we examined the changes in SDF-1 expression and their impacts on the motility of CRC cells. SDF-1 expression was induced in HT-29 and HCT-116 human CRC cell lines through stable clone transfection, which resulted in notable phenotypic changes and altered protein expression levels.

A phase-contrast micrograph analysis revealed significant morphological changes in the HT-29 and HCT-116 colorectal cancer (CRC) cell lines following stable transfection with SDF-1 compared to the control cells (Figure 3A). The transfected cells exhibited notable SDF-1 upregulation, indicating successful transfection. The enzyme-linked immunosorbent assay (ELISA) quantification of SDF-1 protein levels in the conditioned medium from HT-29 and HCT-116 cells cultured separately for 72 h demonstrated significantly elevated SDF-1 levels in the transfected groups (Appendix A). A reverse-transcription–polymerase chain reaction (RT-PCR) performed 48 h post-transfection confirmed the successful expression of SDF-1 mRNA in the transfected cells (Figure 3C). A Western blot analysis showed markedly higher levels of both SDF-1 and CXCR4 in HT-29 and HCT-116 cells treated with recombinant SDF-1 (rSDF-1) at a concentration of 50 µg/mL and those stably transfected with SDF-1 compared to the vehicle-treated controls (Figure 3D). Transwell migration and invasion assays indicated that both rSDF-1 treatment and SDF-1 transfection significantly enhanced the migratory and invasive properties of CRC cells. The addition of AMD3100, a CXCR4 inhibitor, effectively abrogated the migration and invasion induced by SDF-1 (Figure 3E). These data are represented as the mean ± SD from three independent experiments (* *p* < 0.05). A Western blot analysis of epithelial–mesenchymal transition (EMT) markers demonstrated the increased expression of Twist-1, Snail, and Vimentin in CRC cells treated with SDF-1, indicating EMT activation via SDF-1/CXCR4 signaling. The endogenous SDF-1 presence resulted in the upregulation of CXCR4, Vimentin, Twist, and Snail, and the downregulation of E-cadherin. In contrast, treatment with AMD3100 led to the downregulation of CXCR4, Vimentin, Twist, and Snail, and the upregulation of E-cadherin (Figure 3F).

Furthermore, 3D organotypic raft cultures were utilized to show that, compared with the NF group, co-culturing with CAFs significantly increased the percentage of invading cells. A histological examination showed the organotypic tissue invasiveness of HT-29 and HCT-116 cells in various SDF-1 expression groups, including cancer-associated fibroblasts (CAFs), rSDF-1, pMSCV SDF-1/normal fibroblasts (NFs), and pMSCV SDF-1/CAFs. Invasive behavior was observed in all SDF-1-expressing groups, whereas non-invasiveness was noted in NFs and pMSCV SDF-1/CAFs treated with AMD3100 (Figure 3G). These findings were consistent with the observations from the Transwell assays, suggesting that SDF-1 may promote cell migration and invasion via an EMT mechanism. Interestingly, stable clone co-cultures with NFs exhibited a more significant invasion than that of rSDF-1-treated co-cultures with CAFs (Figure 2D). Additionally, SDF-1 upregulated the expression of CXCR4 and further enhanced its effects on CRC cells, suggesting that SDF-1 may exert its migratory functions through both paracrine and autocrine pathways. These results indicate a mechanism whereby EMT, driven by the cell-extrinsic effects of SDF-1, remodels the TME and contributes to the acquisition of aggressive traits in tumor progression associated with the transition in colon cancer. Collectively, these findings underscore the link between SDF-1 expression and the EMT process in CRC cells.

### 3.4. SDF-1 Enhances Spheroid Formation and Promotes Cancer Stemness in CRC Cells

In our study, we meticulously examined the influence of SDF-1 on the stem-like properties of CRC cells, specifically focusing on the roles played by CAFs in this regulatory mechanism. SDF-1, which is secreted by CAFs, was hypothesized to foster an environment conducive to the development and maintenance of CSC characteristics, which are critical for tumor progression and the resistance to conventional therapies. We conducted a series of experiments to determine the impact of SDF-1 on sphere formation—a hallmark of CSCs—in CRC cells. Our results clearly demonstrated that exposure to CAF-CM enriched with SDF-1 notably enhanced the sphere-forming capabilities of CRC cells, indicating an increase in CSC-like properties. These findings are visually represented in Figure 4A, which displays a significant uptick in spheroid formation under SDF-1-stimulated conditions compared with the control. Further, the expression of well-established CSC markers, CD44 and CD133, was robustly upregulated in CRC cells treated with SDF-1. The immunofluorescent staining, which is depicted in Figure 4B, highlights this upregulation, providing visual confirmation of the elevated CSC marker expression induced by SDF-1. Quantitative assessments confirmed that these changes were statistically significant, underscoring the role of SDF-1 in promoting CSC traits.

Moreover, we explored the expression of OCT4, a transcription factor that is essential for sustaining stemness. The Western blot analysis and subsequent quantification revealed that OCT4, along with CD44 and CD133, exhibited markedly higher expression levels in CRC cells subjected to treatments with CAF-CM and SDF-1 transfection compared with those of the control groups. This pattern is illustrated in Figure 4C, which shows that SDF-1 not only sustained but potentially enhanced the stem-like phenotype in CRC cells. Collectively, these observations elucidate the paracrine and autocrine mechanisms by which SDF-1 secreted by CAFs augments the CSC phenotype in CRC cells, contributing to the aggressive tumor behavior and poor patient outcomes. Such insights highlight the potential of targeting the SDF-1/CXCR4 axis as a therapeutic strategy for disrupting the CSC niche within the CRC tumor microenvironment.

### 3.5. SDF-1-CXCR4 Signaling in Tumor Growth through Xenograft Models

Next, we employed tumor xenograft assays to explore the impact of SDF-1 expression on the growth of colon cancer cell xenografts in a nude mouse model. Our detailed investigations revealed a cell-number-dependent relationship with SDF-1 expression, where tumorigenicity was significantly enhanced. Notably, xenografts derived from cells expressing SDF-1 exhibited a marked increase in tumor size compared with the control groups. This difference was statistically significant, with a *p*-value of less than 0.05, confirming the robustness of our findings (Figure 5A,B).

To further elucidate the molecular dynamics underpinning these observations, we utilized immunohistochemical techniques to examine the expression levels of key markers within the tumors. Our analysis revealed that tumors derived from PMSV-SDF-1-transfected cells displayed elevated levels of SDF-1 and its receptor CXCR4, alongside the increased expression of the stemness-associated markers CD44 and OCT4, with moderately enhanced levels of Nanog. In contrast, the control tumors, where SDF-1 expression was not induced, showed significantly lower levels of these markers (Figure 5C,D).

This comprehensive analysis not only substantiates the critical role of SDF-1 in promoting tumor growth but also highlights its influence on the molecular profile of colon cancer xenografts. The clear correlation between SDF-1 expression and the enhanced expression of stemness markers underscores the complex interplay between SDF-1 and its downstream targets. These findings significantly advance our understanding of the mechanisms by which SDF-1 contributes to the pathogenesis of colon cancer and provide a solid basis for future investigations aimed at targeting this pathway as a therapeutic strategy.

### 3.6. Immunohistochemical Staining Reveals the Prognostic Role of CAF-Mediated SDF-1/CXCR4 Signaling and Its Association with Liver Metastases in CRC Patients

In this investigation, we rigorously examined the prognostic impact of SDF-1/CXCR4 signaling mediated by CAFs in CRC through a comprehensive immunohistochemical analysis. Our findings revealed that normal colorectal tissues typically lacked SDF-1 and CXCR4 staining, serving effectively as a negative control (Figure 6A,E). In contrast, as shown with arrows, the CRC tissues in the early (Figure 6B,F) and advanced stages (Figure 6C,G) exhibited intense but variable cytoplasmic and cell membrane staining for SDF-1 and CXCR4, indicating active signaling involvement. 

To evaluate the diagnostic accuracy of the SDF-1 and CXCR4 staining intensity, sensitivity and specificity analyses were performed. The staining intensity was categorized as positive or negative, with intensities of 2 and 3 considered positive and 0 and 1 considered negative. The clinical outcome data were used to classify cases as either positive or negative. A 2 × 2 contingency table was constructed to compare the staining results with the clinical outcomes, enabling the calculation of the sensitivity and specificity. The results of the sensitivity and specificity calculations were as follows: sensitivity—0.92 and specificity—0.53.

These values indicate that the staining intensity of SDF-1 and CXCR4 is highly sensitive (92%) for the correct identification of positive cases and demonstrates fair specificity (53%) for the correct identification of negative cases. This suggests that the staining intensity could represent a reliable diagnostic marker for identifying clinical outcomes related to the SDF-1 and CXCR4 expression in colorectal cancer (Appendix A).

However, in CRC tissues with liver metastasis, SDF-1 was predominantly cytoplasmic-stained, whereas CXCR4 was notably nuclear-stained, suggesting a distinct translocation from the cytoplasm to the nucleus, an unusual pathological phenomenon that is usually correlated with aggressive tumor behavior and metastatic potential (Figure 6D,H). This nuclear localization of CXCR4 in the liver metastasis tissues from CRC patients was markedly more pronounced than that in cells from normal colorectal tissues, underscoring the critical role of the SDF-1/CXCR4 axis in facilitating the metastasis of CRC to the liver (Figure 6F–H). The concurrent expression of SDF-1 and CXCR4 at sites of early or advanced invasive carcinoma and liver metastases highlighted the clinical significance in terms of SDF-1/CXCR4 signaling, acting as a robust predictor of liver metastasis and profoundly affecting the overall prognosis of CRC patients. High-magnification imaging (with a scale bar of 300 μm) enabled the detailed visualization of these expression patterns, providing deep insights into the molecular dynamics within the tumor microenvironment. IHC presentations of the SDF-1/CXCR4 axis from representative patient samples were further examined to test the sensitivity (92%) and specificity (53%) and to validate the usefulness of the current tool (Appendix A). 

## 4. Discussion

CRC is the third most common malignancy worldwide and ranks as the second leading cause of cancer-related mortality. The metastatic dissemination of CRC, particularly to the liver and less frequently to the lungs, significantly contributes to these high mortality rates. A pivotal factor in this process is the presence of CAFs at the invasive front of tumors, as they are strongly associated with a poor prognosis and increased likelihood of recurrence. This correlation highlights the potential of targeting the TME as a strategic approach in the treatment of metastatic CRC. 

The emergence of CAFs within the CRC TME is driven by a dynamic interaction between cancer cells and fibroblasts; this is primarily mediated through the secretion of cytokines, chemokines, and exosomes. This interaction facilitates cellular communication involving the transfer of microRNAs, lncRNAs, proteins, mRNAs, metabolites, and other biologically active molecules [38]. It is generally accepted that resident fibroblasts in colon tissues are the principal progenitors of CAFs in primary CRC. Supporting this, CAFs identified in liver metastases from CRC patients exhibit protein expression profiles similar to those of liver-resident fibroblasts. Furthermore, evidence from studies on human mammary fibroblasts shows that tumor-resident fibroblasts progressively acquire CAF-like characteristics, such as an enhanced α-SMA expression and pro-tumorigenic properties, as the tumor evolves [39,40].

Consistent with these insights, our findings indicate that CAFs isolated from CRC patients’ tissues display an activated phenotype with an increased expression of α-SMA and vimentin, which is indicative of functional and morphological transformations. Moreover, the elevated expression of CD10 and GPR77 in CAFs relative to NFs suggests their pivotal role in the process of differentiation from NFs. This observation aligns with the findings of previous studies, confirming the transformative influence of the TME on fibroblast behavior. These data underscore the complex interplay within the CRC TME and emphasize the transformative role of CAFs in promoting tumor aggressiveness. The findings of this study not only reinforce the significance of CAFs in CRC progression but also validates targeting the stromal components of the TME as a viable therapeutic strategy for inhibiting metastasis and improving patient outcomes [41,42]. 

Furthermore, our study demonstrated that, unlike NFs, CAFs induce EMT and greater aggressiveness in the CRC cell lines HT-29 and HCT-116, highlighting the role of CAFs in facilitating CRC metastasis. The interaction between cancer cells and CAFs, which is mediated through various growth factors and cytokines, including SDF-1, forms a positive feedback loop that may expedite tumor progression. The secretion of transforming growth factor-beta (TGF)-β and SDF-1 by CAFs has been implicated in enhancing the metastatic potential of carcinoma cells undergoing an incomplete EMT. 

Our gene expression microarray analysis further identified a significant upregulation of SDF-1 in CAFs compared with that in NFs. The SDF-1/CXCR4 axis, which is part of the CXC chemokine family, plays a critical role in cancer progression and metastasis across various malignancies, including CRC. Notably, the strong expression of CXCR4 by CRC cells—beyond CAF-mediated SDF-1—correlates with liver metastasis and poorer survival outcomes in CRC patients. Our study fills a gap in the literature by providing molecular and in vivo evidence of the SDF-1/CXCR4 axis’s pivotal role in CRC metastasis and progression. The EMT is recognized as a fundamental process in embryonic development and cancer progression, and it enables epithelial cells to acquire mesenchymal traits [43]. These traits include diminished cell-to-cell adhesion, loss of cell polarity, and enhanced migratory and invasive capabilities. Despite its established importance, the specific contributions of CAF-mediated SDF-1 within the TME and its interaction with the CXCR4 receptor on the surface of tumor cells in CRC has remained insufficiently explored. Our study provides novel insights into this dynamic, demonstrating that the administration of rSDF-1 induces an EMT phenotype in cellular models of CRC. This finding positions SDF-1 as a key mediator in the aggressive behavior of CRC facilitated by the induction of the EMT, highlighting its potential as a target for therapeutic intervention. Through molecular analyses, we confirmed the presence of CXCR4, the primary receptor of SDF-1, which is instrumental in signal transduction processes that drive cancer progression. The chemotactic migration induced by CAF-derived SDF-1 and its interaction with CXCR4 on CRC cells substantiates the operational SDF-1/CXCR4 signaling axis in promoting tumor aggressiveness. 

Furthermore, employing an organotypic culture system allowed us to visualize the infiltration of HT-29 and HCT-116 cells within a matrix layer containing embedded CAFs. This model supports the hypothesis that CAFs, through the secretion of SDF-1, significantly contribute to enhancing the invasive and metastatic potential of CRC cells. These findings underscore the complex interplay between CAFs and CRC cells within the TME—mediated by the SDF-1/CXCR4 axis—as a crucial driver of tumor progression and metastasis. By elucidating these mechanisms, our study not only contributes to a broader understanding of CRC biology but also opens new avenues for the development of targeted therapies aimed at disrupting the detrimental effects of the EMT on cancer progression.

Prior studies have highlighted the roles of TGF-β and SDF-1/CXCL12 in the differentiation of fibroblasts into CAFs and the promotion of a tumor-enhancing phenotype in breast cancer cells. Extending these findings to CRC, our study elucidates the autocrine and paracrine mechanisms of SDF-1/CXCR4 signaling in promoting CRC progression. Our established SDF-1-overexpressing CRC cell lines not only demonstrated morphological changes indicative of the EMT but also exhibited an increased expression of the SDF-1 and CXCR4 genes alongside enhanced invasion and migration capabilities [44]. This study uniquely reveals that CRC cells can augment their sphere-forming abilities and stem cell features through an autocrine SDF-1/CXCR4 signaling pathway, a finding that was previously unreported. This self-reinforcing cycle enhances the CSC properties of CRC (Figure 4), including its drug resistance, sphere-forming ability, invasiveness, and tumor-initiating capacity, potentially leading to relapse [45]. 

To further validate our findings, xenograft models from stable SDF-1-overexpressing CRC clones demonstrated an increased tumor growth ability and SDF-1-CXCR4 signaling overexpression along with pronounced CSC markers. A preliminary clinical validation through an immunohistochemical analysis of CRC patient tissue samples revealed a significant correlation between the expression of SDF-1 or CXCR4 and an unfavorable prognosis—particularly liver metastasis (Figure 6). These results underscore the prognostic potential of SDF-1 in CRC. The tumor stroma—particularly the interaction between CAFs and cancer cells mediated by SDF-1—represents a promising target for therapeutic intervention. Our research has successfully delineated the role of SDF-1 in fostering cancer cell aggressiveness and the underlying mechanisms of CAF–cancer cell interactions. We have demonstrated that the expression of SDF-1 by CAFs directly influences tumor behavior, enhancing aggressiveness through CXCR4 targeting. This interaction triggers a detrimental cycle of SDF-1/CXCR4 signaling, which is correlated with invasive behavior and poor clinical outcomes in CRC patients.

Previous research has demonstrated that the nuclear translocation of CXCR4 is linked with the progression and metastasis of several cancers, including renal cell carcinoma [46,47], non-small-cell lung cancer [48], breast cancer [49], gastric cancer [50], and colorectal cancer [51]. Previous studies have indicated that the nuclear localization of CXCR4 is more prevalent in renal cell carcinoma (RCC) tissues, especially during metastases, and is associated with poor prognosis. The nuclear localization of CXCR4 is mediated by the nuclear localization signal (NLS) gene. The mutation of the NLS gene results in the loss of CXCR4 nuclear localization in RCC cells. Mechanistically, CXCR4 nuclear localization promotes the nuclear accumulation of HIF-1α, which subsequently enhances the expression of HIF-1α downstream genes. Conversely, nuclear HIF-1α promotes the transcription of CXCR4, establishing a feedforward loop [46]. The nuclear localization of CXCR4 is analogous to the mechanisms observed for the epidermal growth factor receptor (EGFR), another membrane receptor. Lin et al. demonstrated that, upon binding with the epidermal growth factor, EGFR translocates to the nucleus, binds to the promoter region of the cyclin D1 gene, and increases its transcription, thereby promoting cell proliferation. Thus, the nuclear translocation of EGFR precedes its role as a transcriptional regulator, leading to observable phenotypic changes [52,53]. Similarly, Wang et al. confirmed that the prolonged exposure to SDF-1 induces the complete nuclear translocation of CXCR4, significantly enhancing invasiveness. This suggests that the nuclear localization of CXCR4 may be involved in transcriptional regulation within the nucleus, impacting genes that control invasion, migration, and chemotaxis [47]. Cell culture experiments have shown both rapid and slow responses of CXCR4 to SDF-1 stimulation. However, our experiments provide limited evidence to support this hypothesis. Future studies should investigate whether nuclear localization requires prolonged and sustained SDF-1 stimulation. Our animal experiments showed that SDF-1-transfected HCT-116 cells exhibited diffuse cytoplasmic, non-nuclear staining. Unless this is a technical artifact, the diffuse cytoplasmic CXCR4 staining represents a third intracellular distribution pattern. Diffuse cytoplasmic CXCR4 staining has been described in the literature [54], but the significance of this pattern has not been fully evaluated. Given the diverse conditions used in various studies, it remains inconclusive whether diffuse cytoplasmic staining is artifactual or represents a true and meaningful pattern. Holland et al. also found differences in CXCR4 activity between metastatic and non-metastatic breast cancer [55], and Wang et al. observed similar results in metastatic RCC [47]. This suggests that our findings in tissue sections may be applicable to other cancers. Further research is needed to elucidate the complexities of this signaling system. If confirmed, the nuclear staining of CXCR4 could serve as a marker to identify metastatic colorectal cancer cells, which could have immediate applications in diagnostic immunohistochemistry. This pattern underscores the crucial role of CXCR4 in not only facilitating cellular communication and migration but also enhancing the metastatic capabilities of tumor cells across diverse cancer types. In our analysis of CRC patients with liver metastases, CXCR4 immunohistochemical staining revealed a notable finding: the nuclei of the metastatic cancer cells displayed a pronounced positivity for CXCR4. This nuclear localization of CXCR4 suggests an enhanced role of this chemokine receptor in the transcriptional regulation mechanisms that drive the progression and metastasis of CRC. Typically, CXCR4 is known for its involvement in cell signaling pathways that influence cell migration and invasion, particularly in the context of cancer metastasis. The nuclear presence of CXCR4 in CRC liver metastasis cells underscores its potential impact not only in cell signaling but also in the gene expression profiles that govern metastatic behavior and tumor aggressiveness. This observation highlights the complexity of CXCR4’s function in metastatic CRC and suggests that its nuclear localization might be associated with advanced disease stages and poorer prognostic outcomes, making it a critical target for therapeutic intervention. 

Collectively, similarly to previous reports, our data also validated that CAF-induced SDF-1 not only promoted the EMT of CRC cells with morphological changes but also helped them obtain subsequent metastatic potential once the whole process triggered the autocrine signaling of SDF-1/CXCR4. According to our primary survey with an immunohistochemical study, the autocrine signaling of SDF-1/CXCR4 in CRC tissues plays a crucial role in locally aggressive behavior and liver metastatic potential. A systemic immunohistochemical and detailed statistical analysis merit further investigation.

The chemokine SDF-1 and its receptor CXCR4 play pivotal roles in the TME, significantly influencing cancer progression and metastasis. The SDF-1/CXCR4 axis is particularly critical in mediating immune cell trafficking and modulating immune responses within the TME. This chemokine–receptor pair is essential not only for the homing and retention of various immune cells, including dendritic cells, T cells, and myeloid-derived suppressor cells (MDSCs) [56], but also for facilitating the escape of cancer cells from immune surveillance. Recent advances in immunotherapy have highlighted the potential of targeting the SDF-1/CXCR4 pathway to enhance anti-tumor immunity. By interrupting this axis, it is possible to disrupt the immunosuppressive network orchestrated by cancer cells and the associated stromal cells, thereby enhancing the efficacy of immune checkpoint inhibitors. For instance, blocking CXCR4 has been shown to reduce the recruitment of MDSCs and regulatory T cells to the tumor site, as these are known to suppress cytotoxic T lymphocytes and hinder effective anti-tumor immune responses [57,58]. Moreover, the inhibition of SDF-1/CXCR4 signaling can potentiate the action of other immunotherapeutic agents by improving the infiltration of effector T cells into the tumor core, a region that is typically characterized by hypoxia and high levels of immunosuppressive cytokines. This enhanced infiltration not only facilitates a more robust anti-tumor response but also helps in remodeling the TME to be more conducive to immune cell activity [59].

Clinical trials utilizing CXCR4 antagonists in combination with established immunotherapies such as PD-1/PD-L1 inhibitors have begun to show promising results. These studies indicate that modulating the SDF-1/CXCR4 axis can lead to improved patient outcomes, especially in cancers where the axis is known to be highly active, such as in melanoma [60] and certain types of leukemia. Conclusively, targeting the SDF-1/CXCR4 signaling pathway represents a promising strategy in the evolving landscape of cancer immunotherapy. By altering the chemotactic signals that govern immune and cancer cell dynamics within the TME, this approach holds the potential to enhance the effectiveness of current immune-based therapies and pave the way for new therapeutic combinations.

Despite the significant findings of this study, several limitations must be acknowledged. Although animal experiments were conducted, the inhibitor was not included in these studies, restricting our understanding of the therapeutic potential and in vivo efficacy of the inhibitor in modulating SDF-1/CXCR4 signaling. Due to resource and time constraints, we prioritized obtaining foundational data from in vitro and baseline in vivo experiments. Comprehensive animal studies with multiple treatment groups, including those treated with the inhibitor, were not feasible within the scope and funding of the current project. In alignment with our commitment to green energy and environmental sustainability (ESG) principles, we extensively utilized tissue culture methods. While this approach minimizes resource use and environmental impact, it may not fully replicate the complex interactions present in a living organism, potentially limiting the generalizability of our findings to in vivo conditions. Additionally, the diffuse cytoplasmic staining of CXCR4 observed in SDF-1-transfected HCT-116 cells could represent a technical artifact rather than a true biological pattern. Further studies are needed to verify this staining pattern and its biological significance.

Our study primarily focused on the short-term responses to SDF-1 stimulation in cell culture, and the potential requirement for prolonged and sustained SDF-1 stimulation to induce the nuclear localization of CXCR4 was not thoroughly investigated. While our findings are significant for colorectal cancer, the applicability of these results to other cancer types remains to be explored. Differences in CXCR4 activity and localization across various cancers suggest that additional research is necessary in order to understand the broader implications of our findings. Lastly, the precise mechanisms by which CXCR4 nuclear localization influences transcriptional regulation were not fully elucidated in this study. Further research is needed to identify the specific genes and pathways affected by nuclear CXCR4. Addressing these limitations in future studies will provide a more comprehensive understanding of the role of SDF-1/CXCR4 signaling in cancer progression and its potential as a therapeutic target.

While the sensitivity and specificity results for SDF-1 and CXCR4 staining intensity are promising (sensitivity = 0.92; specificity = 0.53), several limitations must be considered. In terms of the IHC study, four groups of analyzed data for IHC were considered and each group had a sample size comprising only 10 cases, which may affect the robustness and generalizability of the findings. Additionally, we did not analyze the proportion of stained cells in detail, which could have provided further diagnostic insight. The diffuse cytoplasmic staining that was observed may represent a technical artifact rather than a true biological pattern, requiring further validation. Our analysis was based on a single threshold for staining intensity, and the findings are specific to colorectal cancer, limiting their generalizability to other cancer types. Lastly, the study primarily examined short-term responses to SDF-1 stimulation, and the potential effects of prolonged stimulation were not assessed. Future studies should address these limitations to provide a more comprehensive understanding of the diagnostic value of the SDF-1 and CXCR4 staining intensity.

We conducted animal experiments as part of our study; however, the use of the inhibitor to validate the contrary effect was not included in these experiments. This decision was influenced by several factors:

Initial Study Focus: Our primary objective in the animal experiments was to establish a baseline understanding of the disease model and the effects of SDF-1/CXCR4 signaling without the confounding influence of the inhibitor. This allowed us to characterize the natural progression and pathology more accurately.

Sequential Research Approach: We adopted a sequential approach for our research. The initial phase involved observing the effects of SDF-1/CXCR4 signaling in vivo. Including the inhibitor in this initial phase might have introduced variables that could obscure our understanding of these effects.

Resource Allocation: Conducting comprehensive animal studies with multiple treatment groups, including an inhibitor group, requires significant resources and time. Considering the scope and funding of the current project, we prioritized obtaining foundational data first. Future studies are planned to include the inhibitor to evaluate its therapeutic potential in the established animal model.

Environmental and Sustainability Considerations: In line with our commitment to green energy and environmental sustainability (ESG) principles, we opted to use tissue culture methods to reduce the number of animal experiments. By doing so, we minimized resource use and environmental impact. Our tissue culture results clearly demonstrated the inhibitory effects on cells, supporting the efficacy of the inhibitor.

By focusing on the tissue culture results, we aim to provide robust data that can guide future in vivo studies while adhering to ESG principles. This approach not only ensures responsible resource use but also sets a precedent for future research to consider in vitro models as viable alternatives for drug inhibition studies.

In summary, while the initial animal experiments did not include the inhibitor, this was a deliberate decision to ensure a thorough and systematic investigation. Future research will build on these findings and incorporate the inhibitor to fully assess its potential in vivo, leveraging our current results to promote sustainability in scientific research.

## 5. Conclusions

The expression of SDF-1/CXCR4 in colorectal cancer cells highlights a subset of migratory CSCs that are implicated in the acquisition of an invasive phenotype and metastatic capabilities, as shown in Figure 7. This study substantiates the partial involvement of the EMT in these cells’ progression toward aggressive and metastatic behaviors. Given these findings, SDF-1 emerges as a potential biomarker for CRC, presenting a novel target for therapeutic intervention. Specifically, the blockade of the SDF-1/CXCR4 axis holds promise as a targeted therapeutic strategy for mitigating the metastasis of CRC. To translate these findings from the bench to the bedside, it is crucial that we further investigate the molecular underpinnings of the interactions of CAFs and CRC cells. Sequencing the genomes of CAFs and employing patient-derived tumor explant models will offer invaluable insights into the heterogeneity and dynamics of the TME. Such approaches will not only enhance our understanding of CRC pathophysiology but also pave the way for the development of personalized therapeutic strategies. By targeting the molecular drivers of CRC progression, such as the SDF-1/CXCR4 signaling axis, we can move closer to the effective management and treatment of CRC metastasis. In conclusion, our study underscores the significance of the SDF-1/CXCR4 axis in CRC progression and highlights its potential as a therapeutic target. Further research into the intricate mechanisms of CAF–CRC cell communication and the development of innovative treatment modalities will be instrumental in improving clinical outcomes for CRC patients.

## Figures and Tables

**Figure 1 cells-13-01334-f001:**
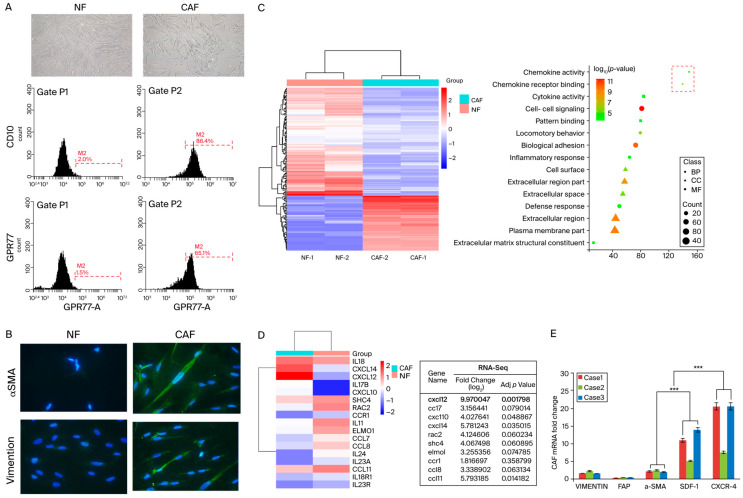
Characterization and functional analysis of CAFs in CRC. (**A**) Flow cytometric analysis illustrating the identification of fibroblasts isolated from samples from CRC patients using the markers CD10 and GPR77, highlighting the heterogeneity within the CAF population. The experiment was conducted in triplicate to ensure the reliability and reproducibility of the results. (**B**) Morphological examination of CAFs through light microscopy, depicting their elongated spindle or stellate shapes with thin cytoplasmic extensions and prominent actin stress fibers. The adjacent panels show the immunofluorescent staining results for α-SMA and vimentin with a comparison of the expression levels in CAFs and NFs, demonstrating significantly higher expression in CAFs; magnification, ×400. (**C**) Gene expression microarray data showcasing significant changes in the expression profiles of the proteins, including chemokines and cytokine receptors, secreted by CAFs compared with NFs, with alterations marked as greater-than-five-fold increases. (**D**) Additional microarray results focusing on distinct clusters of upregulated genes related to tumor invasion and metastasis in CAFs. (**E**) qRT-PCR results displaying elevated mRNA levels of vimentin, fibroblast activation protein, α-SMA, SDF-1, and CXCR4 in CAFs extracted from CRC tissues; these were corroborated by corresponding increases in protein levels, reinforcing the pro-tumorigenic role of these markers in the tumor microenvironment. This experiment was conducted in triplicate to ensure the reliability and reproducibility of the results. (Scale bar: 50 μm). Data are shown as representative images and numerical data are represented as the mean ± SD of each group of cells from three separate experiments. *** *p* < 0.0001.

**Figure 2 cells-13-01334-f002:**
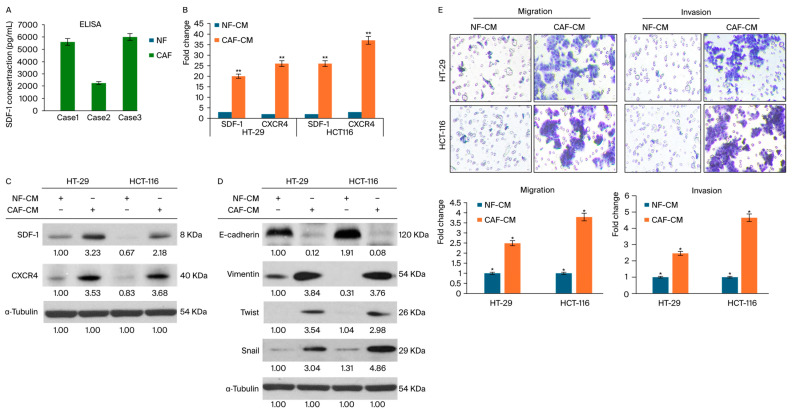
Analysis of SDF-1 and CXCR4 expression in CAFs and NFs, and their impacts on CRC cell motility. (**A**) ELISA results demonstrating the significant upregulation of SDF-1 protein secretion in CAFs compared with other groups, supporting the hypothesis that CAFs enhance SDF-1-induced migration in colorectal cancer. The bar graph shows the quantified protein levels with their standard deviations, illustrating marked differences between the groups. (**B**) qRT-PCR and Western blot analyses depicting the expression levels of SDF-1 and CXCR4 in NFs and CAFs. (**C**,**D**) EMT marker expression via Western blot in CRC cells following treatment with conditioned media. Increased levels of Twist-1, Snail, and Vimentin were observed, indicating EMT activation in response to SDF-1/CXCR4 signaling. (**E**) Migration and invasion were evaluated using Transwell assays to ascertain the influences of different fibroblast types on these processes. Each experiment was conducted in triplicate to ensure the reliability and reproducibility of the results; magnification, ×400. The mean ± SD of each group of cells from three separate experiments is given. * *p* < 0.05. ** *p* < 0.001.

**Figure 3 cells-13-01334-f003:**
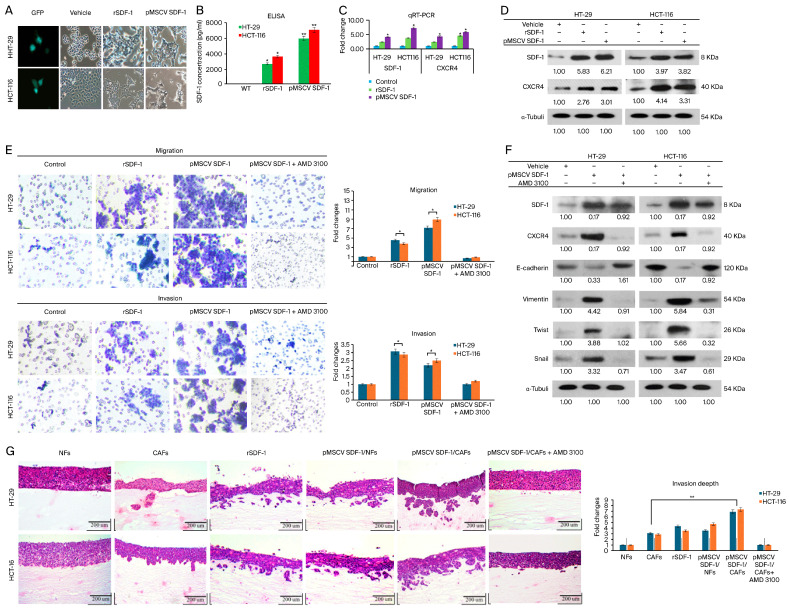
Enhanced migratory and invasive capabilities of CRC cells mediated by SDF-1/CXCR4 signaling. (**A**) Phase-contrast micrograph showing the shape of SDF-1 expression levels in HT-29 and HCT-116 CRC cell lines following stable transfection compared to control cells; magnification, ×400. The data demonstrate significant SDF-1 upregulation in transfected cells, underscoring the effectiveness of the transfection process. (**B**) SDF-1 protein levels in the conditioned medium quantified by ELISA in three groups of HT-29 and HCT116 cells cultured separately for 72 h. (**C**) RT-PCR results detected for 48 h. (**D**) Western blot analysis showed SDF-1 and CXCR4 expression in HT-29 and HCT-116 cells. Compared with the vehicle, higher SDF-1 and CXCR4 expression were observed in rSDF-1 and transfected cells. (**E**) Transwell analysis showed that rSDF-1 and transfected cells demonstrated increased cell migration and invasiveness. AMD3100 inhibition of CXCR4 eliminated the migration and invasion induced by SDF-1; magnification, ×200. Data are represented within images and numerical data are represented as the mean ± SD for each group of cells from three separate experiments. * *p* < 0.05. (**F**) Western blot analysis showed that, in the presence of endogenous SDF-1, CXCR4, vimentin, Twist, and Snail were all upregulated, while E-cadherin was downregulated. Expression of CXCR4, vimentin, Twist, and Snail were downregulated in cells cultured via AMD3100 inhibition, while E-cad expression was upregulated. (**G**) Histological analysis shows organotypic tissue invasiveness of HT-29 and HCT-116 cells in SDF-1 expression groups (CAFs, rSDF-1, pMSCV SDF-1/NFs, and pMSCV SDF-1/CAFs), while NFs and pMSCV SDF-1/CAFs + AMD3100 demonstrated non-invasion. Data are represented as images and numerical data are represented as the mean ± SD for each group of cells from three separate experiments. * *p* < 0.05; ** *p* < 0.01.

**Figure 4 cells-13-01334-f004:**
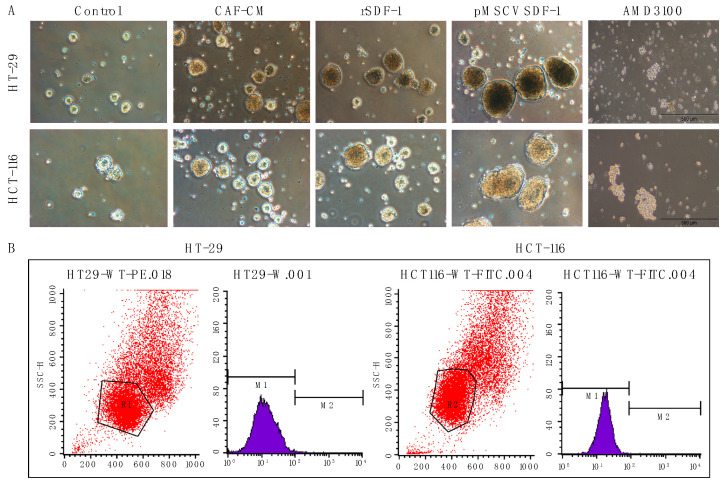
Influence of SDF-1 on CSC characteristics in CRC cells mediated by cancer-associated fibroblasts (CAFs). (**A**) Analysis of sphere formation under different treatment conditions demonstrating the effect of SDF-1 on promoting spheroid phenotypes in CRC cells. This panel shows increased sphere formation in cultures treated with CAF-CM and rSDF-1 in comparison with the control groups. The graph quantifies the sphere formation efficiency, illustrating a significant enhancement in conditions stimulated by SDF-1. (**B**) Expression levels of the CSC markers CD44 and CD133 in CRC cells following exposure to SDF-1. The images of immunofluorescent staining highlight the upregulation of these markers in cells treated with SDF-1, which is indicative of enriched CSC properties. The quantitative analysis below these images shows the relative expression levels of CD44 and CD133, with bars representing statistical significance. (**C**) Detailed expression analysis of the transcription factor OCT4 and CSC markers CD44 and CD133 in CRC cells subjected to various treatments. The Western blot images and corresponding densitometric analysis illustrate that cells exposed to CAF-CM and SDF-1 transfection exhibit substantially higher levels of these proteins compared with the controls, confirming the role of SDF-1 in maintaining CSC characteristics. The graph adjacent to the blots quantifies protein expression, emphasizing the differential impact of SDF-1 on CSC marker expression.

**Figure 5 cells-13-01334-f005:**
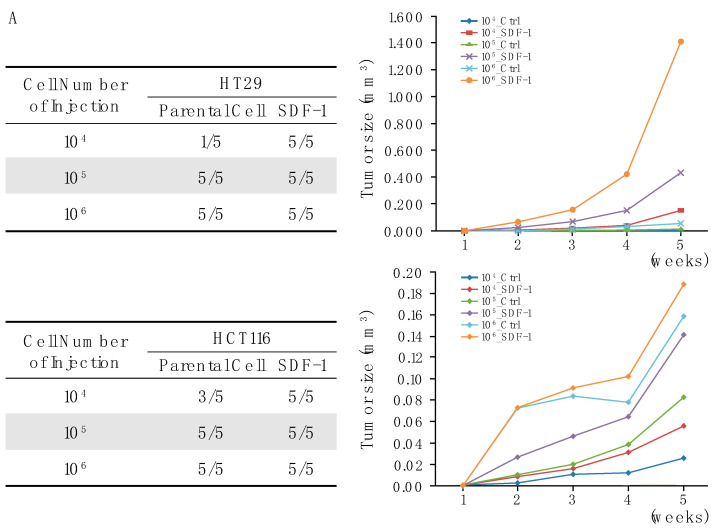
Enhanced tumor growth and molecular characterization in SDF-1-expressing colon cancer xenografts. (**A**) A closer view of the cellular morphology in tumors, with images showcasing differences in cell density and organization between the SDF-1-expressing and control groups and further illustrating the impacts of SDF-1 on tumor architecture and cellular behavior. Quantitative analysis of tumor growth in nude mice xenografted with colon cancer cells. The graph displays a significant increase in tumor size in mice injected with cells overexpressing SDF-1 compared with the control group, demonstrating the role of SDF-1 in promoting tumor growth. Statistical significance with *p*-values of less than 0.05 is indicated. (**B**) Representative images of tumors harvested from the xenograft models, showing the visual difference in tumor size between the SDF-1-expressing group and the controls. (**C**) Immunohistochemical staining for SDF-1, CXCR4, CD44, and OCT4 within the tumor tissues. The images reveal higher expression levels of these markers in the SDF-1-expressing tumors compared with the controls, highlighting the molecular changes associated with SDF-1 expression. (**D**) Analysis of stemness-associated markers in tumor tissues with the quantification of Nanog expression levels. The elevated Nanog levels in SDF-1-expressing tumors suggest an increase in stem-like properties, corresponding to the aggressive nature of these tumors.

**Figure 6 cells-13-01334-f006:**
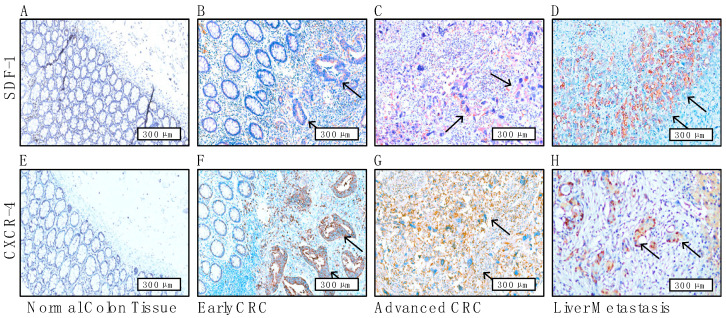
Differential expression of the SDF-1/CXCR4 axis initially from CAF-induced paracrine SDF-1 and ultimately reinforcing SDF-1/CXCR4 signaling in CRC tissues in an autocrine manner, highlighting their roles in tumor progression and metastasis. (**A**,**E**) Immunohistochemical staining images showing the absent expression of SDF-1 and CXCR4 in normal colorectal tissues as negative controls. These panels establish the baseline expression levels for comparison with cancerous tissues. (**B**,**F**) High-magnification images displaying slight cytoplasmic and membrane staining of SDF-1 and CXCR4 in early CRC cell nests and moderate expression (**C**,**G**) in advanced CRC tissues, illustrating the active involvement of the expression of SDF-1 and CXCR4 signaling in CRC tissues and reinforcing the consistency of SDF-1 and CXCR4 expression in these aggressive cancer forms. Lastly, the images in (**D**,**H**) depict strong cytoplasmic positivity for SDF-1 and pronounced nuclear positivity for CXCR4 in CRC patients with liver metastasis. The arrows indicate patterns that demonstrate and reinforce how SDF-1/CXCR4 signaling is closely correlated with tumor behavior and metastatic potential, providing visual evidence of the significant role of the SDF-1/CXCR4 axis in promoting the aggressiveness of CRC. All images have a scale bar of 300 μm.

**Figure 7 cells-13-01334-f007:**
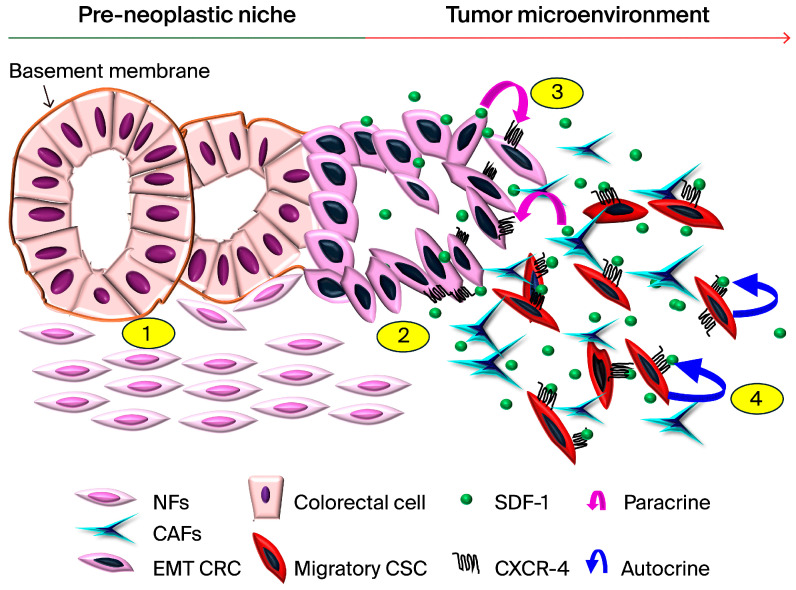
The progression of CRC aggressiveness through CAF-mediated SDF-1/CXCR4 signaling involving both paracrine and autocrine mechanisms; we designed a diagrammatic representation that highlights the following key stages: 1. Hemostasis/Pre-Neoplastic Stage: Illustration of the normal stroma with widely distributed NFs existing adjacent to normal or preneoplastic colonic cells, indicating minimal interaction and a stable dynamic balance between NFs and CRC cells. 2. Transformation into CRC: Depiction of the transition from preneoplastic colonic cells to CRC cells and the EMT with malignant behavior. This stage shows CAFs that have transformed from NFs accumulating around and closely interacting in the invasive fronts of CRC cell nests. The release of SDF-1 by CAFs and its binding to the CXCR4 receptor on CRC cells are highlighted. 3. Paracrine SDF-1/CXCR4 Signaling: Visualization of SDF-1 being secreted by CAFs in a paracrine manner, binding to CXCR4 receptors on CRC cells, and, thus, promoting increased aggressiveness in CRC cells. 4. Autocrine SDF-1/CXCR4 Signaling: Illustration of CRC cells expressing and releasing excessive SDF-1, which binds to their own CXCR4 receptors, creating a self-reinforcing autocrine cycle. This cycle enhances CSC characteristics, aggressiveness, and tumorigenicity, leading to a poor prognosis.

## Data Availability

The data used in this study are available upon request from the corresponding authors.

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
