# Peer review of "Cancer-Associated-Fibroblast-Mediated Paracrine and Autocrine SDF-1/CXCR4 Signaling Promotes Stemness and Aggressiveness of Colorectal Cancers"

_cells, 2024, doi:10.3390/cells13161334_

Round 1

Reviewer 1 Report

Comments and Suggestions for Authors

Summery

This study investigates the role of cancer-associated fibroblasts (CAFs) in promoting colorectal cancer (CRC) progression through the secretion of stromal-derived factor-1 (SDF-1) and its interaction with C-X-C motif chemokine receptor 4 (CXCR4). It demonstrates that SDF-1 enhances CRC cell migration, invasion, and cancer stem cell (CSC) characteristics via paracrine and autocrine signaling. The author shows that inhibiting CXCR4 with AMD3100 can reverse these effects.

Positives

   - The study focusing cancer-associated fibroblasts (CAFs) tumour interaction. This is highly relevant as it can uncover novel therapeutic targets.

   - The study shows a CAF-tumour interaction mechanism, which will create the opportunity for further research. (for example the role of TGFbeta in this process, or the effect on ROCK signalling induced invadopodia and matrix remoddeling, to name a few options). 

   - The study uses different types of techniques to prove the mechanism, which makes it more reliable. 

   - The experiments are well-designed, with multiple independent replicates and appropriate controls

- the paper is well written and easily readable. The discussion is extensive and critical in nature. 

Overall, very nice paper. One obvious improvement would be adding in vivo data for the inhibitors. But you have very nicely explained your rationale for not doing this, and fully agree. We need to do as little animal experiments as possible.

I would suggest to strengthen the paper on two fronts.

- One is the limited number of patients and limited patient data. Improving this would really make this more clinically relevant. So try to increase the number of patients and perform a survival analyses comparing the groups with high and low progression. Or if no survival data is available, include a table with for example percentages and stages.

-The second is that all experiments are based on the same two cell lines. Potentially, mutational status can affect the fibroblasts-tumor interaction. So including more cell lines in 1 or 2 key experiments (certainly not all, it is a proof of principle) would show that your effect is not by chance only in these two celllines.  HT29 is SMAD4 neg, APC mut, p53 mut. HCT116 is SMAD4 pos, p53 WT, APC mut. 

Reviewer 2 Report

Comments and Suggestions for Authors

In the manuscript entitled "Cancer-Associated-Fibroblast-Mediated Paracrine and Autocrine SDF-1/CXCR4 Signaling Promotes Stemness and Aggressiveness of Colorectal Cancers," the authors investigate the role of CAF-derived stromal-derived factor-1 (SDF-1) and its interaction with C-X-C motif chemokine receptor 4 (CXCR4) in colorectal cancer (CRC). The study demonstrates that SDF-1 significantly enhances CRC migration, invasion, and cancer stem cell (CSC) characteristics, such as spheroid formation. Subsequently, authors have performed immunohistochemical analysis to show close relationship between SDF-1 / CXCR4 expression in tumor tissue and patient outcomes. In general, I found this paper to be well-written and on an important and relevant topic. However, I have some comments and suggestions for the author's consideration which are as follows:

1.       In section 3.2, the authors wrote: “We also performed qRT-PCR and a Western blot analysis to examine the expression levels of SDF-1 and CXCR4 in both NFs and CAFs. The qRT-PCR results revealed that the mRNA and protein levels of SDF-1 and CXCR4 were significantly lower in the NF group than in the CAF group, as shown in Figure 2B.” However, Figure 2B lacks such data. The authors also need to correct the figure legend for Figure 2B.

2.       In Figure 3B, the authors measured SDF-1 protein levels in the conditioned medium following cell culture. In one group, the authors treated the cells with rSDF-1 for 72 hours and then measured it in the same media. Therefore, the amount of SDF-1 in these culture media is the recombinant form rather than the secreted one. The authors also need to mention the amount of rSDF-1 used in this experiment.

3.      Furthermore, the authors have not demonstrated that CAF-conditioned media induces SDF-1 secretion in cancer cells, as most of the data presented is either qPCR or Western blot analysis on cell lysates, which primarily represents intracellular SDF-1 rather than the secreted form.

4.      In figure 3C, Bar legends are wrong. Please correct it.

Round 2

Reviewer 2 Report

Comments and Suggestions for Authors

The authors have responded to the reviewers' comments satisfactorily.